# Photoluminescence Dependance of 2-Bromo-3-aminobenzo[*de*]anthracene-7-one on Solvent Polarity for Potential Applications in Color-Tunable Optoelectronics

**DOI:** 10.3390/molecules30132677

**Published:** 2025-06-20

**Authors:** Emmanuel Karungani, Elena Kirilova, Liga Avotina, Aleksandrs Puckins, Sergejs Osipovs, Titus Ochodo, Mildred Airo, Francis Otieno

**Affiliations:** 1Department of Physics and Materials Science, Maseno University, Kisumu 333-40105, Kenya; titusochodo@gmail.com (T.O.); frankotienoo@maseno.ac.ke (F.O.); 2Department of Applied Chemistry, Institute of Life Sciences and Technology, Daugavpils University, LV-5401 Daugavpils, Latvia; aleksandrs.puckins@du.lv (A.P.); sergejs.osipovs@du.lv (S.O.); 3Institute of Chemical Physics, University of Latvia, 1 Jelgavas Street, LV-1004 Riga, Latvia; 4Department of Chemistry, Maseno University, Kisumu 333-40105, Kenya; mildredairo@gmail.com

**Keywords:** benzanthrone, 2-bromo-3-aminobenzanthrone, optoelectronics, fluorescence, OLEDs, ICT

## Abstract

The novel benzanthrone derivative, 2-bromo-3-aminobenzo[*de*]anthracene-7-one (2-Br-3-NH_2_BA), was synthesized and extensively characterized to investigate its photophysical behavior in various solvents. It was prepared through selective bromination of 3-aminobenzanthrone using N-bromosuccinimide in dimethylformamide at −20 °C. Featuring a donor–π–acceptor (D–π–A) structure, 2-Br-3-NH_2_BA exhibits pronounced solvatochromism due to the intramolecular charge transfer (ICT) between the amino donor and the carbonyl acceptor groups. Optical measurements conducted in eight solvents of varying polarity revealed a significant bathochromic shift in both absorption and fluorescence emission, with emission maxima red-shifting by over 110 nm from non-polar to polar environments. Corresponding reductions in the optical band gap energies, as calculated from Tauc plots, further support solvent-induced electronic state modulation. Additionally, quantum yield analysis showed higher fluorescence efficiency in non-polar solvents, while polar solvents induced twisted intramolecular charge transfer (TICT), leading to emission quenching. These findings demonstrate the sensitivity of 2-Br-3-NH_2_BA to environmental polarity, making it a promising candidate for color-tunable luminescent applications in optoelectronics and sensing. However, further studies in the solid state are required to validate its applicability in device architectures such as OLEDs.

## 1. Introduction

Color-tunable materials responsive to environmental factors, such as polarity, are at the forefront of optoelectronic research. Among these, organic light emitting diodes (OLEDs) have emerged as a leading technology in modern display and lighting applications offering advantages such as high efficiency, thinness, flexibility, and the capability for multicolor emission [1,2,3,4,5]. A crucial component in OLEDs is the emissive layer (EML), usually regarded as the heart of OLEDs, which determines the device’s color output, quantum efficiency, and overall performance [6,7]. To meet the growing demand for next-generation optoelectronics, there is a strong interest in developing novel organic materials with tunable photoluminescence properties [8].

The evolution of OLED materials has progressed from first-generation fluorescent materials too second-generation phosphorescent materials, and further to third generation thermally activated delayed fluorescence (TADF) materials [9,10]. While first generation materials, such as tris (8-hydroxyquinolinato) aluminum (Alq_3_), have seen commercial success, they suffer from poor color purity and emitted brightness [6]. Phosphorescent emitters, although highly efficient, depend on expensive rare metals, and often show stability issues under operational stress leading to a shorter device lifetime [11,12]. TADF materials offer a metal-free alternative capable of achieving 100% internal quantum efficiency by harvesting both singlet and triplet excitons [9,13]. However, they frequently require complex molecular design to achieve high efficiency, and may suffer from efficiency roll-off at high brightness levels [14]. Additionally, most of these materials are restricted in their ability to offer tunable emissions without structural modifications.

Benzanthrone derivatives have attracted attention in this regard due to their robust photostability, extended π-conjugation, strong fluorescence, and tunable electronic properties [15,16,17,18]. These molecules are especially valued for their capacity to undergo functionalization, which can modulate their electronic characteristics and photophysical behavior. They can be chemically modified to introduce donor (D) and acceptor (A) moieties onto the chromophore [19,20] forming a donor–π–acceptor architecture which promote intramolecular charge transfer (ICT) [21]. The ICT-architecture molecules often display solvatochromism behavior, wherein their emission characteristics shift in response to the polarity of the surrounding medium. Notably, the positioning of the electron-donating and electron-withdrawing groups on the benzanthrone core can drastically influence the extent of the ICT, and thus impact their performance in real-world applications [22,23], especially in developing color-tunable optoelectronic materials that can be tailored during processing or operation.

Recent studies on benzanthrone derivatives have revealed that such systems exhibit environment-sensitive fluorescence, high thermal stability, and a large Stokes shift. For instance, Schiff bases derived from 3-aminobenzanthrone and heterocyclic aldehydes exhibit solvent-dependent fluorescence behavior, showing negative fluorosolvatochromism in their imine form and enhanced, positive solvatochromism upon reduction to their corresponding amines. These reduced derivatives not only display more intense luminescence but demonstrate greater thermal stability, with decomposition temperatures reaching up to 300 °C [24]. Similarly, amidine-functionalized benzanthrone dyes show bright yellow-to-orange fluorescence and significant Stokes shifts, with emission characteristics sensitive to solvent polarity, indicating strong intramolecular charge transfer processes; additionally, quantum chemical calculations have confirmed planar aromatic structures and possible amidine orientations that support bathochromic shifts [25]. Collectively, these findings highlight the structural versatility of benzanthrone dyes and underscore the importance of rational molecular design in optimizing their optical performance for targeted applications.

However, despite the structural versatility and electronic adaptability of the benzanthrone scaffolds, their potential applications in OLEDs remains underexplored. To the authors knowledge, there is no existing literature on these derivatives’ applications in OLEDs.

In this study, we focus on 2-bromo-3-aminobenzo[*de*]anthracene-7-one (2-Br-3-NH_2_BA), a novel benzanthrone derivative featuring an amino group at the 3-position and a bromine atom at the 2-position. The amino group serves as an electron donor, while the bromine acts as an electron-withdrawing substituent, together establishing a D–π–A architecture that is highly conducive to the ICT. Such a structural configuration is anticipated to exhibit solvent-sensitive photophysical behavior, including solvatochromism and emission modulation, both of which are critical for optoelectronic utility.

We present here a comprehensive analysis of the synthesis, structural features, and solvent-dependent optical properties of 2-Br-3-NH_2_BA. The compound’s behavior was investigated across a series of solvents with varying polarities to elucidate its absorption and emission characteristics, quantum yields, Stokes shifts, and dipole moment variations. These studies aim to assess the suitability of 2-Br-3-NH_2_BA for applications in optoelectronic devices, particular for color-tunable devices, such as OLEDs and biosensors. By understanding the fundamental photophysical behavior of this compound, we can better evaluate its role not only in color-tunable applications but in advancing the next generation of organic semiconducting materials.

## 2. Results and Discussions

### 2.1. Synthesis of 2-Bromo-3-aminobenzo[de]anthracene-7-one

The successful synthesis and structural elucidation of functionalized benzanthrone derivatives form a critical foundation for exploring their optoelectronic and semiconducting potential. In this study, 2-bromo-3-aminobenzanthrone was synthesized as a key intermediate, offering dual reactivity through its electron-rich amino group and electrophilic bromo substituent. These functionalities not only enable subsequent donor–acceptor modifications but influence the molecule’s electronic distribution and intermolecular interactions. Comprehensive characterization using spectroscopic and structural techniques was employed to confirm the molecular identity, purity, and crystalline nature of the compound.

For the synthesis, we modified the method of Takamura-Enya et al. [26] by using solutions of the starting materials, thereby increasing the yield of the target product. The 2-Br-3-NH_2_BA compound was obtained through bromination of 3-aminobenzanthrone (3-NH_2_BA) using N-bromo-succinimide (NBS). NBS is used because it is a highly specific bromination agent both in free-radical substitutions and the electrophilic addition of unsaturated systems as it releases small quantities of bromine to enhance effective bromination [27], additionally DMF was used as a solvent, since bromination of aromatic derivatives with NBS using DMF gives high levels of para sensitivity [27]. During the reaction, the temperature was maintained at −20 °C in the freezer to prevent the immediate bromination of 3-NH_2_BA. After the disappearance of the initial amine spot, water was added to the reaction mixture to react with excess NBS and also to precipitate the product. The synthesis mechanism is summarized in Figure 1.

### 2.2. Structural and Elemental Properties of 3-NH_2_BA and 2-Br-3-NH_2_BA

The functional groups of 2-Br-3-NH_2_BA were identified using FTIR. The FTIR spectra of 2-Br-3-NH_2_BA are shown in Figure 2.

The obtained spectroscopic data fully confirms the structure of 2-Br-3-NH_2_BA. The IR spectra exhibit an absorption band at 3447 cm^−1^ and 3357 cm^−1^ which is due to N-H stretching, indicating the presence of -NH_2_ in the synthesized 2-Br-3-NH_2_BA. The 3 peaks in the 3-NH_2_BA spectrum suggests splitting due to intermolecular hydrogen bonding, possibly caused by hydrogen bonding environments involving the amine group and the surrounding molecular structure. By contrast, 2-Br-3-NH_2_BA shows only two prominent peaks in this range, which may be due to the electronic effects of the bromine substituent which influences the N-H stretching frequencies by altering the electron density around the amine group. The peak at 3236 cm^−1^ is due to C-H stretching, a characteristic peak for an aromatic group. The strong peak observed at 1654 cm^−1^ corresponds to C=O, the characteristic absorption band at 1570 cm^−1^ is as a result of C=C stretching which is a characteristic peak for the π-conjugated medium. The strong peak at 659 cm^−1^ is assigned to C-Br stretching which indicates the presence of bromine. The absence of a C-Br peak in 3-NH_2_BA and its presence in our compound implies that we successfully synthesized our target compound (2-Br-3-NH_2_BA). Additionally, these spectral differences support successful structural modification during synthesis.

Furthermore, the MS measurements of 3-NH_2_BA and 2-Br-3-NH_2_BA (Figure 3a) were performed to confirm chemical composition of the samples. 

The major peaks at 322/325 matches m + 1 enhance the molecular ion 324.16 as expected. The two peaks (*m*/*z* 322/325) were identified as bromine, which corresponds to its two isotopes (79 Br and 81 Br) [28]. This doublet (two peaks separated by two atomic mass units) of similar intensity near the molecular region indicates that fragmentation starts with the loss of bromine. The peak at 243 corresponds to 3-NH_2_BA after the loss of Br 81. Compared to the MS of the initial compound 3-NH_2_BA (Figure 3b), it is evident that bromination was successfully performed to obtain the compound.

### 2.3. Effect of Solvents on the Photo-Physical Properties of 2-Bromo-3-aminobenzanthrone

The spectroscopic properties of the synthesized 2-bromo-3-aminobenzanthrone were evaluated and the corresponding data were summarized in Table 1 below. The absorption and emission spectra were recorded in eight organic solvents with different polarities. The absorption and emission spectra of 2-Br-3-NH_2_BA are shown in Figure 4.

For the synthesized 2-Br-3-NH_2_BA the absorption and emission properties are caused by charge transfer between the electron donating -NH_2_ at the 3-C and the electron accepting carbonyl group on the benzanthrone chromophore system through the π conjugated part of the benzanthrone molecule [24,25,29]. 

The 2-Br-3-NH_2_BA compound exhibits two absorption bands, one at 360–380 nm which are ascribed to π-π*, and the second, intense broad band at 444–500 nm are the ICT from the donor to acceptor through the π-bridge [30]. The results which are in agreement with the already existing literature on benzanthrone derivatives, like those synthesized by Maļeckis et al. [31], and it was also observed in the benzanthrone derivatives synthesized by Orlova et al. [24].

The maximum absorption of 2-Br-3-NH_2_BA shows a bathochromic shift of 56 nm with an increase in solvent polarity from 444 nm in non-polar hexane to 500 in ethanol, as shown in Figure 4a. This observation is in agreement with the existing literature [32,33,34]. The bathochromic shift is indicative of an enhanced intramolecular charge transfer (ICT) in the polar solvents, where the excited state is more effectively stabilized. This stabilization leads to a reduced gap between HOMO and LUMO levels, a critical requirement for optoelectronics’ efficiency. Therefore, this property suggests that 2-Br-3-NH_2_BA can absorb light more efficiently across a broader spectral range when processed with suitable solvents. The broadening of the absorption bands in the polar media also implies a higher degree of delocalization of the π-electron system, which is beneficial for efficient photon capture in solar cell applications. The effect of solvent polarity is further illustrated in Figure 5.

Unlike the solvent dependent absorption spectra, the emission of 2-Br-3-NH_2_BA shows a pronounced solvatochromic bathochromic shift with an increase in solvent polarity, as shown in Figure 4b. It shows an intense fluorescence emission corresponding to the 0-0 band at 508 nm in hexane and, as the solvent polarity increases, the fluorescence emission peak shows a bathochromic shift of about 115 nm. The sudden increase in the emission spectra of ethanol is due to the fact that the excited state is more stabilized by the intermolecular hydrogen bonding present in ethanol. The larger observed bathochromic shift in the emission spectra relative to the corresponding absorption spectra indicates that there is higher stabilization of the excited state dipole moments than the ground state dipole moment and the fluorescence emission is, as a result of the ICT mechanism which occurs during excitation from the strong electron, donating -NH_2_ to the electron withdrawing carbonyl group.

The difference in absorption and emission behavior can be related to how the energy states are affected by the solvent during the π–π* electronic transition. Therefore, the lack of solvent effect on the absorption solvatochromism is a result of unequal stabilization of both the ground state and the excited state by the different solutions used. This phenomenon was also observed and explained in 2018 by Kirilova et al. They explained that, after excitation, the re-ordering of the solvent molecules surrounding the luminophore leads to a more stabilized excited state with lower energy [29]. Therefore, 2-Br-3-NH_2_BA is more polar in the excited than the ground state. Such stabilization is essential for promoting efficient exciton generation and separation.

### 2.4. Full Width Half Maxima (FWHM)

The full width half maxima (FWHM) of the fluorescence emission peaks were analyzed to provide insights into the vibronic coupling and the degree of structural relaxation in the excited state [35]. The obtained values in Table 1 show an increase in the FWHM with an increase in solvent polarity. This broadening of the emission bands in the polar solvents is a characteristic of stronger ICT states, which typically leads to less structured emissions due to increased vibrational relaxation pathways. The broader FWHM in DMSO, DMF, and EtOH supports the idea of more delocalized states and stronger solute–solvent interactions.

For optoelectronic applications, such broad emissions suggest the potential for effective overlap with the solar spectrum, which is essential in maximizing photon harvesting and light utilization. Since the obtained values lie within the typical range (70–120 nm) reported for polymers and small-molecular donors on organic optoelectronic systems [36,37], it suggests that 2-Br-3-NH_2_BA is an ideal candidate for use in color-tunable applications.

### 2.5. Effect of Solvent Polarity on Stokes Shift (∆ύ) of 2-Br-3-NH_2_BA

The Stokes shift (∆ύ) was calculated in wavenumber (unit = cm^−1^) according to the following equation:(1)$$∆ύ=(\frac{1}{\lambda_{Abs}}-\frac{1}{\lambda_{Em}})\times 10^{7}$$

From Table 1, the ∆ύ values increased significantly with solvent polarity, from 83 nm in hexane to 153 nm in ethanol, which is also clearly illustrated in Figure 6. This is because more polar solvents cause higher stabilization of the excited state than the ground sate due to dipole–dipole interactions, and also implies that there is an increase in the dipole moment upon excitation [38,39]. The 2-Br-3-NH_2_BA compound exhibits large Stokes shifts due to the ICT caused by its donor–acceptor π-conjugated system. 

Such high Stokes shifts indicate substantial reorganization of the molecular geometry upon excitation, and support the ICT mechanism [40]. Large Stokes shifts may reflect the efficient stabilization of charge-separated states, potentially reducing energy losses during exciton dissociation. The shifts also suggest the molecule’s suitability for light-harvesting in a broad spectral region.

### 2.6. Fluorescence Quantum Yield (ϕ)

Using the comparative method, the $\phi_{f}$ was determined by using Equation (2) where in our case, P8:CHCl_3_ was used as a reference. (2)$$\phi_{fs}=\phi_{fs}(\frac{A_{r}}{A_{s}})(\frac{E_{s}}{E_{r}})(\frac{n_{s}}{n_{s}}{)}^{2}$$(3)$$\phi_{fs}=\phi_{fs}(\frac{m_{s}}{m_{r}})(\frac{n_{s}}{n_{s}}{)}^{2}$$(4)$$\phi_{fs}=\phi_{fs}(\frac{m_{s}}{m_{r}})$$
where $\phi_{f}$ is the fluorescence quantum yield, m is gradient of the plot of integrated fluorescence intensity against absorbance, n is the refractive index of the solvent, A is the absorbance of the dye, and E is the integrated fluorescence of the emitted light. The subscripts ‘r’ and ‘s’ refer to the reference and test sample fluorophore, respectively.

However, by combining the ER and EA terms (as well as As and Es terms), since their ratio is equal to the calibration line produced when plotting the integrated fluorescent intensity against the absorbance of the solution, Equation (2) can be simplified to Equation (3). However, if the solvent used for both the reference and test sample is the same, the value of $(\frac{n_{s}}{n_{s}}{)}^{2}$ is equal to 1, hence the new form of Equation (2) becomes Equation (4); therefore $\phi_{f}$ of the test sample is calculated from the product of the $\phi_{f}$ of the reference and the quotient of the two gradients [41,42]. The values of E were calculated using Origin 2021 software.

From Table 1 and Figure 7a, it can be observed that ϕ decreases with an increase in ∆ύ. Figure 7b also shows that the fluorescence quantum yield also decreases as the solvent polarity increases. This is because, in polar solvents with high hydrophilicity, the dye’s molecular conformation is probably twisted, leading to the twisted intramolecular charge transfer (TICT) which weakens the emission of the fluorophore [43,44]. Therefore, the observed weakening of the quantum yield with the increasing solvent polarity is due to TICT processes.

### 2.7. Estimation of 2-Br-3-NH_2_BA Dipole Moments

The estimation of the ground and excited state dipole moments in various solvents with varied dielectric constants (ϵ) and refractive index (n) can be achieved using Equations (5)–(7), known as the Lippert–Mataga [45], Bakhshiev’s [46] and the Kawski–Chamma–Viallet [47] equations, respectively.(5)$$∆ύ=m_{1}k_{1}\left(\epsilon ,n\right)+Constant$$(6)$$∆ύ=m_{2}k_{2}\left(\epsilon ,n\right)+Constant$$(7)$$\frac{\lambda_{abs}+\lambda_{ems}}{2}={-m}_{3}k_{3}\left(\epsilon ,n\right)+Constant$$
Further, the Lippert–Mataga, Bakhshiev, and Kawski–Chamma–Viallet polarity functions were determined according to the following equations:(8)$$k_{1}\left(\epsilon ,n\right)=\left[\frac{\epsilon -1}{2\epsilon +1}-\frac{n^{2}-1}{2n^{2}+1}\right]$$(9)$$k_{2}\left(\epsilon ,n\right)=\left[\frac{\epsilon -1}{\epsilon +2}-\frac{n^{2}-1}{n^{2}+2}\right]\frac{\left(2n^{2}+1\right)}{\left(n^{2}+2\right)}$$(10)$$k_{3}\left(\epsilon ,n\right)=\frac{\left(2n^{2}+1\right)}{\left({2n}^{2}+2\right)}\left[\frac{\epsilon -1}{\epsilon +2}-\frac{n^{2}-1}{n^{2}+2}\right]+\frac{3(n^{4}-1)}{2(n^{2}+2{)}^{2}}$$

The polarity function values, k_1_, k_2_, and k_3_ were obtained using Equations (8)–(10), respectively, and are tabulated in Table 2.

The obtained values of k1,k2, and k3 are further presented graphically in Figure 8.

Figure 8 takes the form of an equation of a straight line (y=mx+C) with slopes $m_{1}$, $m_{2}$, and $m_{3}$, respectively, which can be represented as follows:(11)$$m_{1}=\frac{2(\mu_{e}-\mu_{g}{)}^{2}}{hca^{3}}$$(12)$$m_{2}=\frac{2(\mu_{e}-\mu_{g}{)}^{2}}{hca^{3}}$$(13)$$m_{3}=\frac{2({\mu_{e}}^{2}-{\mu_{g}}^{2})}{hca^{3}}$$
where $\mu_{g}$ and $\mu_{e}$ are the ground and excited state dipole moments of the fluorophore, respectively, h is the Planck’s constant, c is the speed of light in a vacuum, and a is the Onsager radius calculated by the Edward volume addition method [48].$$a={\left(\frac{3M}{4\pi \delta N}\right)}^{1/3}$$
where M is the molecular weight of 2-Br-3-NH_2_BA (324.16 g/mol), δ is the density obtained by Chemsketch software (1.642±0.06 cm^3^), and N is the Avogadro constant. We obtained a=4.29Å.

From the above plots, which were fitted to a straight line, we obtained the values of $m_{1}$, $m_{2}$, and $m_{3}$ (for eight data points), from which we calculated the $\mu_{g}$, $\mu_{e}$, and ∆μ, and recorded in Table 3.



$$1Debye=3.33364\times 10^{-31}cm=10^{-18}esu\;cm$$



If the excited and ground state are parallel with the unchanged molecular symmetry upon electron excitation, we can now explicitly obtain $\mu_{g}$ and $\mu_{e}$ from Equations (11)–(13) [38] according to the following equations:(14)$$\mu_{g}=\frac{m_{3}-m_{2}}{2}\sqrt{\frac{hca^{3}}{2m_{2}}}$$(15)$$\mu_{e}=\frac{m_{2}+m_{3}}{2}\sqrt{\frac{hca^{3}}{2m_{2}}}$$

For ($m_{3}>m_{2})$, $\mu_{e}$ takes the following form:(16)$$\mu_{e}=\left[\frac{m_{2}+m_{3}}{m_{3}-m_{2}}\right]$$

However, if they are not parallel but form an angle, then Equations (14) and (15) takes the new form, as follows:(17)$$\cos\theta =\frac{1}{2\mu_{g}\mu_{e}}\left[\left(\mu_{g}^{2}+\mu_{e}^{2}\right)-\frac{m_{2}}{m_{3}}(\mu_{e}^{2}-\mu_{g}^{2})\right]$$

By rewriting Equations (11)–(13) we can easily obtain the change in the dipole moment as follows:(18)$$m=\frac{{2∆\mu }^{2}}{hca^{3}}$$
where ∆μ is the change in dipole moments.

The calculated values reveal that the excited state dipole moments are greater than those of the ground state, indicating that the exited state is more polar than the ground state [38]. The enhanced polarity accounts for the large solvatochromic shifts observed, suggesting a significant redistribution of electron density upon excitation due to charge transfer processes.

### 2.8. Effect of Solvent Polarity on Molar Extinction (ε) Coefficient

The molar extinction coefficient (ε) quantifies the intrinsic ability of the molecule to absorb light at a specific wavelength [49]. It was obtained from the absorption data according to the Beer–Lambert equation [50], and is graphically represented in Figure 9.(19)λ=εcl
where λ and l are the absorption wavelength and the path length (usually 1 cm), respectively, c is the sample concentration, and ε is the molar extinction coefficient.

The analysis of ε across the studied solvents revealed that polar solvents, such as DMSO and DMF, significantly enhanced the absorption capacity of 2-Br-3-NH_2_BA. This improvement can be attributed to better solvation and molecular alignment, which increase the transition dipole moment. 

Across the different solvent environments, ε increased with solvent polarity, with the highest values recorded in EtOH, DMSO, and DMF. This increase correlates with the enhanced ICT nature of the molecule in the polar media. In the context of OSCs, a higher extinction coefficient implies a greater probability of photon absorption [30], which is essential for generating sufficient charge carriers in the active layer. Additionally, it ensures that more incident photons are absorbed and converted into excitons, thereby improving the overall efficiency of the device. This property implies that 2-Br-3-NH_2_BA can also be a promising candidate in OSCs.

### 2.9. Effect of Solvents on the Optical Band Gap of 2-Br-3-NH_2_BA

The optical bandgap (Eg), is a critical parameter for selecting materials in optoelectronic applications, as it represents the minimum amount of energy required to excite an electron from the valence to the conduction band [51]. The Eg of 2-Br-3-NH_2_BA was determined using Tauc plots, where $(ah\nu {)}^{2}$ was plotted against photon energy (hν), as shown in Figure 10.

The results indicated a solvent-dependent decrease in Eg from 2.204 eV in hexane to 1.967 eV in DMSO (Table 4). 

In 2-Br-3-NH_2_BA, the amino group acts as an electron donor and the brominated carbonyl moiety acts as an electron acceptor. Upon excitation, electron density is transferred from donor to acceptor creating a more polar excited state [52]. In polar solvents, this excited state is more stabilized, leading to a redshift in both the absorption and emission spectra, and resulting in a smaller bandgap [53], as shown by the Tauc plots in Figure 10 and Table 4. The bandgap narrows from 2.204 eV in non polar hexane to 2.002 eV in polar DMSO [54]. However, the narrowest bandgap, 1.967 eV, is observed in ethanol. This is because, in addition to ethanol’s polarity, the excited state is more stabilized by the intramolecular hydrogen bonding present in ethanol [55]. This trend reflects the stabilization of the LUMO level in polar solvents, and supports the enhanced ICT behavior.

While perovskite quantum dots (PQDs) and carbon dots (CDs) have emerged as strong contenders in optoelectronic applications, they come with certain limitations. PQDs exhibit outstanding optical characteristics, including high quantum yield and narrow emissions. However, their poor stability at ambient temperature, susceptibility to degradation, and toxicity due to the presence of lead raises environmental and health concerns [56,57]. On the other hand, CDs are cheap, biocompatible, and highly stable, but typically have lower photoluminescence quantum yields and low color purity, which limits their applications in high-resolution displays [58]. By contrast, 2-Br-3-NH_2_BA presents a metal-free alternative with a well-defined donor–π–acceptor structure that offers clear solvatochromic shifts and large Stokes shifts with enhanced ICT. The compound’s emission profile is sensitive to solvent environment and polarity, enabling its potential utility in color-tunable platforms. Although this study focuses on solution-phase characterization, the well-structured and chemically stable aromatic scaffold of 2-Br-3-NH_2_BA offers synthetic flexibility, and may provide advantages in terms of processability and long-term thermal stability over some QD-based systems. Future comparisons involving thin-film optical quality and electroluminescence efficiency will be essential to position this molecule within the broader optoelectronic material landscape.

## 3. Materials and Methods

### 3.1. Materials and Basic Measurements

All the reagents used were of analytical grade purchased from Sigma Aldrich (Munich, Germany), and were used as received. The reaction progress was monitored using thin layer chromatography (TLC) on silica gel plates (Fluka F60 25 4 (Honeywell, Charlotte, NC, USA), 20 × 10, 0.2 mm) in a solvent system of benzene–acetonitrile (*v*/*v* 3:1) under UV light visualization using F15T8 BLB Black UV Lamp CE RoHS 365nm (Haihong, Ningbo, China). Purification of the products was caried out using column chromatography on Merck Kieselgel (230–240 mesh) with dichloromethane, as the eluent melting point was determined on the MP 70 Melting Point System and are not corrected.

The identification of the chemical bonds was performed by means of Fourier-transform infrared (FTIR) spectrometry. A Bruker Vertex 70v vacuum spectrometer (Bruker Corporation, Billerica, MA, USA) equipped with an attenuated total reflection (ATR) accessory was used in this study. 

Mass spectrometry (MS) was used to identify the molecules present. The Shimadzu GCMS-QP2010 system (Shimadzu Corporation, Kyoto, Japan) was used for the analysis. The gas chromatograph was equipped with an electronically controlled split/splitless injection port and a 5% diphenyl-/95% dimethylpolysiloxane fused-silica capillary column (Rtx-5SIL-MS, 30 m × 0.32 mm, 0.25 μm film thickness). Mass spectra (MS) were registered by a mass spectrometer in electron ionization mode (ionization energy of 70 eV). Detection was realized in the scanning mode within the range of *m*/*z* 35–500.

The spectral properties of the investigated compound were measured in eight solvents, including hexane, benzene, chloroform, ethyl acetate, acetone, ethanol, dimethylformamide, and dimethyl sulfoxide, with concentrations of 10^−5^ M at an ambient temperature in 3.5 mL quartz cuvettes. All solvents were of p.a. or analytical grade. The fluorescence emission spectra were recorded on a FLSP920 (Edinburgh Instruments Ltd., Edinburgh, UK) spectrofluorometer in the visible range 450–800 nm and the absorption spectra were obtained using the UV–visible spectrophotometer CamSpec M550 (Spectronic CamSpec Ltd., Leeds, UK).

### 3.2. Synthesis of 2-Bromo-3-aminobenzanthrone

The 2-bromo-3-aminobenzanthrone compound was synthesized using the modified method of Takamura-Enya et al. [26], where 0.38 g of N-bromosuccinimide (NBS) was dissolved in 5 mL of dimethylformamide (DMF) and added with mixing by portion (1–2 mL) at −20 °C to a solution of 0.5 g of 3-amino-benzanthrone in 7 mL DMF for 20–30 min. After the disappearance of the initial amine spot, 20–30 mL water was added to the reaction mixture and allowed to precipitate. The product was obtained after filtration, followed by washing with 2–3 mL ethanol as a dark red powder with a yield of 96% and an m.p. of 242–243 °C. FTIR: 3447, 3357, 3236, 1654, 1570, 1095, 766, 659; MS: *m*/*z*: 326(22), 323/325[m^+^], 324(23), 243(6), 217(25), 189(27), 108(20), 94(23); ^1^H NMR ([D6]DMSO): d = 8.84 (d, J = 8.8 Hz, 1H), 8.70 (s, 1H), 8.64 (dd, J = 7.6, 0.8 Hz, 1H), 8.45 (d, J = 8.0 Hz, 1H), 8.26 (dd, J = 8.0, 0.8 Hz, 1H), 7.82 (t, J = 8.0 Hz, 1 H), 7.74 (dt, J = 8.0, 0.1 Hz, 1 H), 7.47 (t, J = 7.9 Hz, 1H), 6.88 ppm (s, 2H); ^13^C NMR ([D6]DMSO): d = 181.6, 147.9,134.3, 133.1, 130.5, 130.4, 130.3, 130.2, 130.2, 127.7, 127.6, 127.3, 126.0, 125.2, 125.2, 124.0, 112.2 ppm [26].

## 4. Conclusions

In summary, we have successfully synthesized and characterized 2-Br-3-NH_2_BA, a benzanthrone-based fluorophore exhibiting strong solvent-dependent photophysical properties. The compound demonstrates pronounced solvatochromism, large Stokes shifts, and variable fluorescence quantum yields across solvents of different polarity. These effects are attributed to intramolecular charge transfer and excited-state stabilization in polar environments, resulting in a significant modulation of the optical band gap. While these findings suggest potential utility in color-tunable luminescent applications, it is important to acknowledge that all measurements were performed in a dilute solution. This study does not include solid-state or thin-film analyses, which are critical for evaluating performance in real optoelectronic devices, such as OLEDs. Future work will therefore focus on fabricating and characterizing thin films of 2-Br-3-NH_2_BA and integrating them into prototype devices to assess their practical suitability.

## Figures and Tables

**Figure 1 molecules-30-02677-f001:**
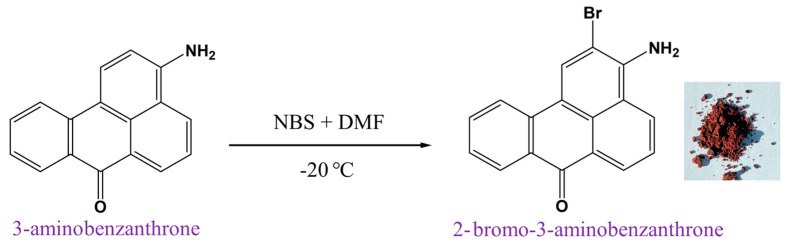
Schematic representation of the synthesis procedure of 2-bromo-3-aminobenzathrone using 3-amino-benzathrone, as the precursor.

**Figure 2 molecules-30-02677-f002:**
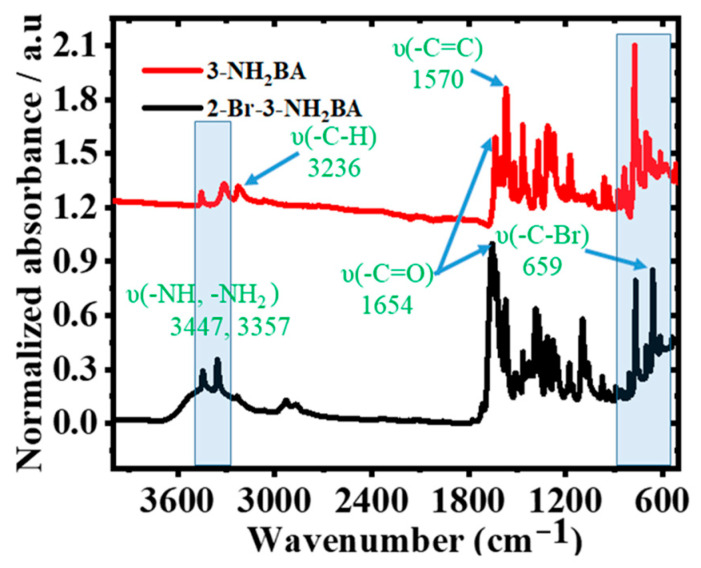
FTIR spectra of 3-NH_2_BA and 2-Br-3-NH2BA.

**Figure 3 molecules-30-02677-f003:**
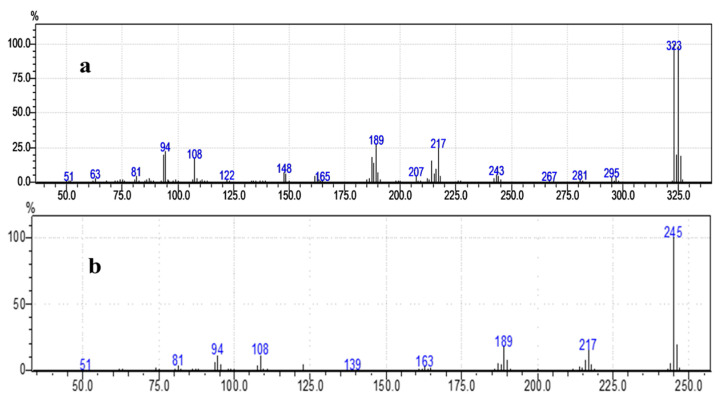
Mass spectrometry spectra of (**a**) 2-Br-3-NH_2_BA and (**b**) 3-NH_2_BA.

**Figure 4 molecules-30-02677-f004:**
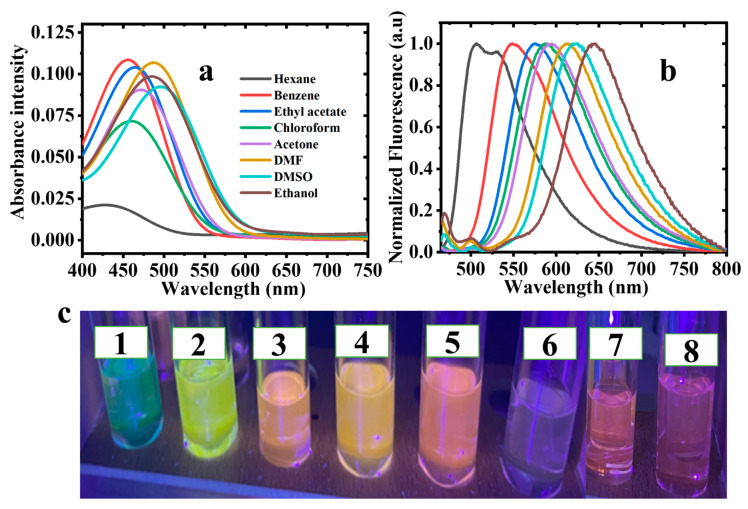
Effect of solvents on the (**a**) absorption and (**b**) emission spectra of 2-Br-3-NH_2_BA. (**c**) Picture of 2-Br-3-NH_2_BA in the presence of (1) hexane, (2) benzene, (3) chloroform, (4) ethyl acetate, (5) acetone, (6) ethanol, (7) DMF, and (8) DMSO after UV lamp illumination at 365 nm.

**Figure 5 molecules-30-02677-f005:**
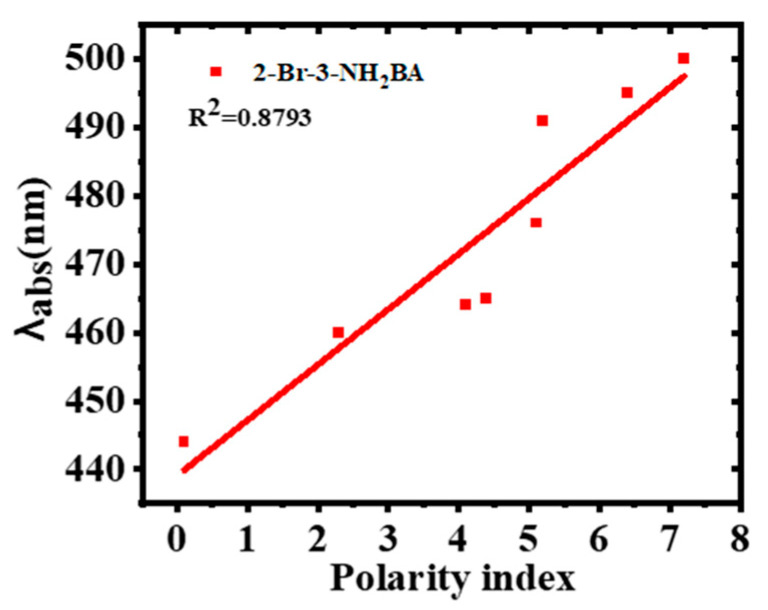
Variation of maximum absorption of 2-Br-3-NH_2_BA with the solvent polarity index.

**Figure 6 molecules-30-02677-f006:**
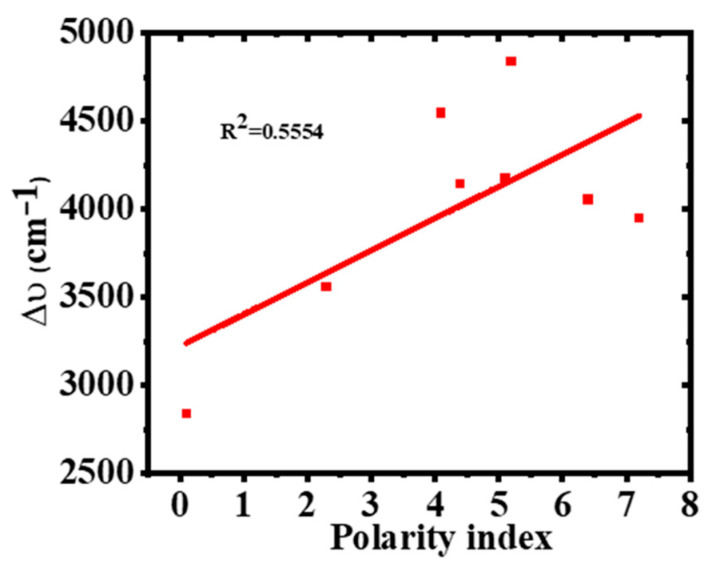
Effect of solvent polarity index on Stokes shift.

**Figure 7 molecules-30-02677-f007:**
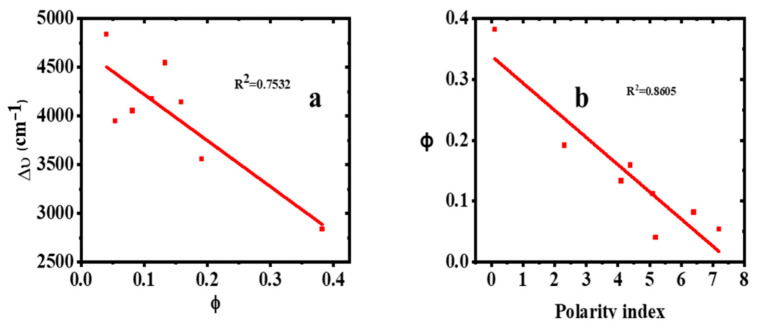
Variation of (**a**) Stokes shift with ϕ and (**b**) ϕ with polarity index.

**Figure 8 molecules-30-02677-f008:**
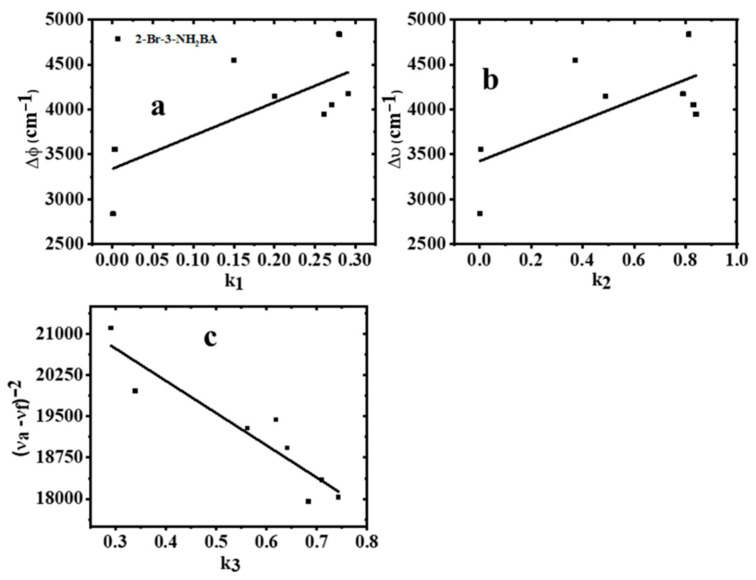
Variation of Stokes shift with (**a**) the Lippert–Mataga polarity function, (**b**) the Bakhshiev polarity function, and (**c**) the Kawski–Chamma–Viallet polarity function.

**Figure 9 molecules-30-02677-f009:**
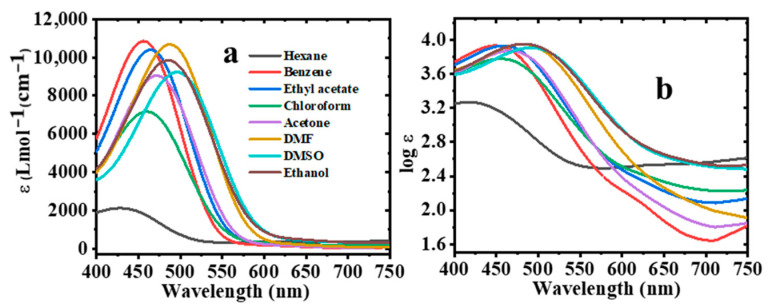
Effect of solvent polarity on (**a**) absorption coefficient and (**b**) extinction coefficient.

**Figure 10 molecules-30-02677-f010:**
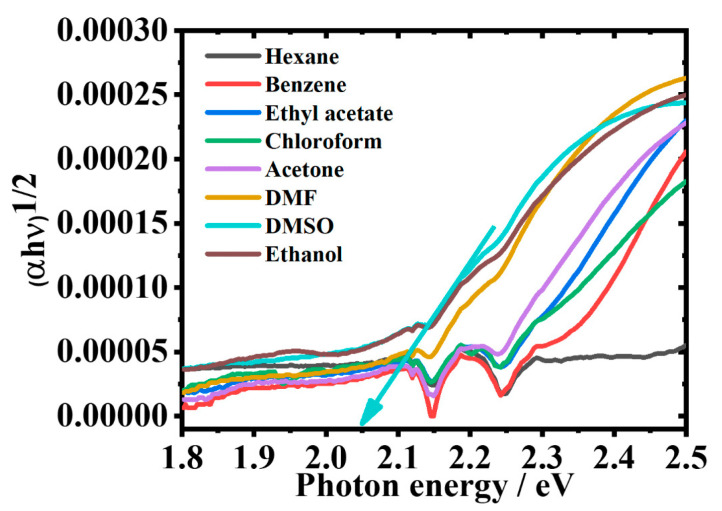
Tauc plots of 2-Br-3-NH_2_BA in different solvents.

**Table 1 molecules-30-02677-t001:** Photophysical properties of 2-Br-3-NH_2_BA; maximum absorption (λabs), logarithm of molar extinction coefficient (log ε), maximum fluorescence (λems), polarity index (P^I^), Stokes shift (∆ύ), full width half maxima (FWHM), and Quantum yield (ϕ).

Solvent	λabs (nm)	log ε	λems (nm)	P^I^	∆ύ (cm^−1^)	FWHM (nm)	ϕ
Hexane	444	3.415	508	0.1	2837.48	86	0.382160861
Benzene	460	4.110	550	2.3	3557.31	89	0.191602503
Chloroform	464	3.915	588	4.1	4544.92	91	0.133138514
Ethyl acetate	465	4.077	576	4.4	4144.27	92	0.158607877
Acetone	476	4.012	594	5.1	4173.39	94	0.112330369
Ethanol	491	4.040	644	5.2	4838.65	95	0.040469667
DMF	491	4.082	613	6.4	4053.39	97	0.081548666
DMSO	500	4.015	623	7.2	3948.64	98	0.054056066

**Table 2 molecules-30-02677-t002:** Dielectric constant, refractive index, and polarity function values of different solvent.

Solvent	n	ϵ	k_1_	k_2_	k_3_
Hexane	1.375	1.89	0.001	0.0015	0.290818
Benzene	1.501	2.274	0.003	0.0045	0.3391
Chloroform	1.443	4.807	0.15	0.3722	0.562419
EtOAc	1.372	6.02	0.2	0.4893	0.619282
Aceton	1.359	20.7	0.291	0.7903	0.6419
Ethanol	1.361	24.55	0.28	0.8131	0.6841
DMF	1.446	36.71	0.271	0.8307	0.7105
DMSO	1.478	46.6	0.261	0.8407	0.7432

**Table 3 molecules-30-02677-t003:** Values of $m_{1}$,
$m_{2}$,
$m_{3}$,
$\mu_{g}$,
$\mu_{e}$, and ∆μ.

2-Br-3-NH_2_BA	$m_{1}$	$m_{2}$	$m_{3}$	$\mu_{g}$	$\mu_{e}$	∆μ
Slopes	3699	1135	−5860	5.89	8.72	2.83
R^2^	0.571	0.553	0.867			

**Table 4 molecules-30-02677-t004:** Effect of solvent polarity on Eg
of 2-Br-3-NH_2_BA.

Solvent	Hexane	Benzene	Chloroform	Ethyl Acetate	Acetone	Ethanol	DMF	DMSO
Band gap (eV)	2.204	2.198	2.175	2.161	2.148	1.967	2.051	2.002

## Data Availability

The data presented in this study are available in this article.

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
