# Peer review of "Photoluminescence Dependance of 2-Bromo-3-aminobenzo[de]anthracene-7-one on Solvent Polarity for Potential Applications in Color-Tunable Optoelectronics"

_molecules, 2025, doi:10.3390/molecules30132677_

Round 1

Reviewer 1 Report

Comments and Suggestions for Authors

This paper can be considered after major revisions. My comments are given below,

  1. How 2-Bromo-3-amino 2 benzo[de]anthracene-7-one can tune the color change properties? Is it solvent effect? Band gap tuning can be discussed with respect to the change in color change. Is particle size affected by the change in color properties?
  2. Page 4, Figure 2, please mention and label the FTIR peaks.
  3. Which solvent is best to get the narrow band gap?
  4. It lacks data for Organic Light Emitting Diodes
  5. What are the features of any materials to explored for Organic Light Emitting Diodes? Is carbon dots can be used? How is reported material is better than perovskite quantum dots or carbon dots for LED?
  6. Compare your results with carbon dots and perovskite quantum dots based LED.
  7. Abstract should be rewritten. 
  8. Conclusion should mention the limitations. 

Author Response

Comments 1: How 2-Bromo-3-amino 2 benzo[de]anthracene-7-one can tune the color change properties? Is it solvent effect? Band gap tuning can be discussed with respect to the change in color change. Is particle size affected by the change in color properties?

Response 1: In 2-Br-3-NH2BA, the amino group and the brominated carbonyl moiety acts as an electron donor and the brominated carbonyl moiety acts as an electron and acceptor respectively. Upon excitation, electron density is transferred from donor to acceptor creating a more polar excited state (1). In polar solvents, this excited state is more stabilized leading to a redshift in both absorption and emission spectra resulting in a smaller bandgap (2) as shown by the Tauc plots in figure 10 and table 4. The bandgap narrows from 2.204 eV in non-polar hexane to 2.002 eV in polar DMSO. (3). Thank you for pointing this out. We agree with this comment. Therefore, we have included a clear explanation of this mechanism in the manuscript on page 13paragraph 1 line 379-388

Comments 2: Page 4, Figure 2, please mention and label the FTIR peaks.

Response 2: Thank you for pointing this out. We have updated figure 2 to include clearly labelled FTIR peaks and their discussion in the corresponding section (page 4) which explicitly refers to key vibrational modes which are N-H, C-H, C=O, C=C and C-Br stretches on lines 142, 149, 150 and 151 respectively.

Comments 3: Which solvent is best to get the narrow band gap?

Response 3: The narrowest bandgap, 1.967 eV is observed in ethanol, this is because, in addition to ethanol’s polarity, the excited state is more stabilized by intramolecular hydrogen bonding present in ethanol (4). We agree with this comment. In the manuscript, we now explicitly state that ethanol due to its high polarity and intramolecular hydrogen bonding highly stabilizes the LUMO leading to a narrow bandgap page 13 lines 385-387.

Comments 4: It lacks data for Organic Light Emitting Diodes

Response 4: We acknowledge this limitation and now clearly state in the revised abstract and conclusion (pages 1 and 14 respectively) that this study explores the optical properties in solution and not device-level performance. The discussion emphasizes on potential application in color tunable optoelectronics. The discussion of OLEDs is now framed as a potential future direction and not as a demonstration of working device.

Comments 5: What are the features of any materials to explored for Organic Light Emitting Diodes? Is carbon dots can be used? How is reported material is better than perovskite quantum dots or carbon dots for LED?

Response 5: While perovskite quantum dots (PQDs) and carbon dots (CDs) have emerged as strong contenders in optoelectronic applications, they come with certain limitations. PQDs exhibit outstanding optical characteristics including high quantum yield and narrow emission. However, their poor stability at ambient temperature, susceptibility to degradation and toxicity due to presence of lead raises environmental and health concerns (5,6). On the other hand, CDs are cheap, biocompatible and highly stable, but typically have lower photoluminescence quantum yields and low color purity which limits their applications in high-resolution displays (7). In contrast, 2-Br-3-NHâ‚‚BA presents a metal-free alternative with a well-defined donor–π–acceptor structure that offers clear solvatochromic shifts and large Stokes shifts. The compound’s emission profile is sensitive to solvent environment and polarity, enabling its potential utility in color-tunable platforms. Although this study focuses on solution-phase characterization, the well-structured and chemically stable aromatic scaffold of 2-Br-3-NHâ‚‚BA offers synthetic flexibility, and may provide advantages in terms of processability and long-term thermal stability over some QD-based systems. Future comparisons involving thin-film optical quality and electroluminescence efficiency will be essential to position this molecule within the broader optoelectronic material landscape. We agree with this comment and a result, we have now added a new paragraph in the discussion (page 13, paragraph 2 lines 390-406) outlining the key desirable features for LED materials. We also now compare 2-Br-3-NH2BA to carbon quantum dots, and Perovskite quantum dots. Relevant literature references are also used for this comparison [52-58].

Comments 6: Compare your results with carbon dots and perovskite quantum dots based LED.

Response 6: Thank you for pointing this out. We have added a new comparative discussion to the manuscript (page 13 paragraph 2) highlighting the similarities and distinctions between our compound and other well-established emissive materials such as carbon dots (CDs) and perovskite quantum dots (PQDs).

Comment 7: Abstract should be rewritten. 

Response 7: Thank you for your recommendation. We have now re-written the abstract for clarity and precision. It now avoids overstatements about OLED applications and focuses more on the observed solvent-dependent photophysical phenomena of 2-Br-3-NH2BA.

Comment 8: Conclusion should mention the limitations. 

Response 8: We agree with this comment. We have revised the conclusion section to mention that the current study is limited to solution-phase properties. We also state that further studies, including solid state thin film analysis and device fabrication are required to validate performance in OLED.

Reviewer 2 Report

Comments and Suggestions for Authors

This study synthesizes and systematically characterizes a novel benzanthrone derivative, 2-Br-3-NHâ‚‚BA, with a focus on investigating its solvent-dependent photophysical properties and its potential application in OLED emissive layers. The experimental design is rigorous and the data are convictive. The paper innovatively reveals the strong solvatochromic coloration effect and intramolecular charge transfer (ICT) characteristics of the compound. I recommendation this paper to be major revision.

Comments:

1.There are a lot of format and writing issues : such as subscripts of "2-Br-3-NHâ‚‚BA" and ”NH2“, line 294, and line 297.

2. Abbreviations not defined: The full names of ICT, FWHM, etc. should be provided when they first appear and using the abbreviations after that.

3. The solvent fluorescence photograph must be labeled with the UV wavelength (e.g., 365 nm) and FWHM unit missing (nm).

4. There are No OLED device data at all, I think it is not accepted.

5. Photoluminescence Dependance of 2-Br-3-NHâ‚‚BA on Solvent Polarity is only for solution, how can apply the color changes into OLED? OLED is solid.

6. Fig 2: X axe is without labels.

Author Response

Comment 1: There are a lot of format and writing issues: such as subscripts of "2-Br-3-NHâ‚‚BA" and ”NH2“, line 294, and line 297.

Response 1: The authors have taken a keen look into the formatting and have now carried out formatting, including the proper use of subscripts and chemical formulas thought the manuscript.

Comments 2: Abbreviations not defined: The full names of ICT, FWHM, etc. should be provided when they first appear and using the abbreviations after that.

Response 2: Thank you for these observations. We have now defined all abbreviations such as ICT (Intramolecular charge transfer) line 68-69 and FWHM (full width half maxima) line 177-178 upon first mention and abbreviated thereafter.

Comment 3: The solvent fluorescence photograph must be labeled with the UV wavelength (e.g., 365 nm) and FWHM unit missing (nm).

Response 3: For illumination of 2-Br-3-NH2BA in different solvents, we used F15T8 BLB Black UV Lamp CE RoHS 365nm. We agree. Therefore, we have added the UV lamp wavelength (365nm) in the caption of figure 4C and included units for FWHM (nm) in table 1.

Comment 4: There are No OLED device data at all, I think it is not accepted.

Response 4: The authors humbly acknowledges that in deed the study did not include fabrication of OLED devices nor simulations and have now revised manuscript by removing direct link of OLED performance and have also clearly indicated that this study focuses on photophysical behavior of 2-Br-3-NH2-BA in solution.

Comment 5: Photoluminescence Dependance of 2-Br-3-NHâ‚‚BA on Solvent Polarity is only for solution, how can apply the color changes into OLED? OLED is solid.

Response 5: This is a valid point. We have clarified in the abstract, discussion and conclusion that although solvatochromism is demonstrated in solution, the insights on ICT, quantum yield, Stokes shift and excited state polarity provides useful design parameter for solid state applications. This includes thin films which can be embedded in matrices.

Comment 6: Fig 2: X axe is without labels.

Response 6: This was an omission and we have now updated Figure 2 to include proper labelling of the x-axis (wavenumber in cm-1)

Reviewer 3 Report

Comments and Suggestions for Authors

This work from Karungani et al. describes the luminescence behaviour of 2-Bromo-3-aminobenzo[de]anthracene-7-one in solution. The synthesis of this derivative and its thorough characterization are reported. Furthermore, eight different solvents with different polarities have been used in order to evaluate the dependence of several luminescence properties on the environment: absorption and emission maxima, full width half maxima, Stokes shift, fluorescence quantum yield, dipole moment, molar extinction coefficient and optical band gap, all differ considerably depending on the solvents, due to the molecular architecture.

It is my opinion that this work is really well written and that it could be of interest to the journal readership, but there are few points to be addressed. I suggest acceptance of this work after minor revisions. Below some specific notes:

  1. In every 2-Bromo-3-aminobenzo[de]anthracene-7-one, “de” should be written in italic.
  2. Line 49: I think that “immitted” could be “emitted”.
  3. Line 50: “they” is a repetition of the subject.
  4. In general, in the whole work it is often mentioned that this compound should be proper as emissive layer in an OLED device. While the luminescence properties described are surely interesting and worth of noting, it is my personal opinion that they are not sufficient to evaluate the potential use in OLEDs. Emissive species in such devices are not in solution, but rather in thin films, often embedded in matrices. Therefore, evaluation of the luminescence properties in films, or at least in solid state is necessary before furhter conclusions can be drawn.
    I suggest to remove or at least reducing the mentions to the using of this species in OLED devices, especially in the title, which could sound misleading as one could expect to find the evaluation of a device performances in this work.
  5. Lines 115-116: there are too many repetitions of the word “method”.
  6. If possible, I would suggest to record 1H and 13C NMR spectra of 2-Br-3-NH2BA, to further complete the characterization of this compound.
  7. Line 157: I think it should be “m/z 322/325” and not “m/z 322/3250”
  8. For the seek of completeness, I would suggest to add the solvent polarity indexes in Table 1. Also, in the caption, there is no reference to log ε, while all the other entries are described.
  9. Whenever a linear regression is shown (such as in figure 5, 6, 7 and 8), I would add the R2.
  10. For Figure 6, 7 (especially 7a), 8a, 8b, I would be cautious to use a linear fitting, as the points seem not to follow a linear relation. I would consider it more a general trend, rather than a linear relation.
  11. I several points, it is mentioned that OLEDs functions upon absorbing light (lines 249-250 and 350-351). While the light absorption properties are important to understand the photophysical behaviour of systems like this, I think that this is not correct, as in OLEDs the excitons are created upon applying a voltage to the systems. OLEDs converts electricity into light, while Solar Cells convert light into electricity. I may be missing the point of such statements, but I think they are wrong or, at least, misleading.
  12. Figure 7b: I think it could be better to invert x and y axis, as the quantum yields is considered the dependent variable and the solvent polarity the independent one.
  13. Lines 309-310: there is an opening bracket, but not a closing one.
  14. Lines 361-363: the sentence lacks a main preposition.

Author Response

Comment 1: Accept. The English could be improved to more clearly express the research.

Response 1: The authors sincerely thank the reviewer and have also thoroughly edited the manuscript for grammar, clarity, and technical precision as advised.

Round 2

Reviewer 1 Report

Comments and Suggestions for Authors

Can be published now

Reviewer 2 Report

Comments and Suggestions for Authors

The OLED part has been revised properly, it is acceptable now.